**Subject Category:**
Biology (whole organism)

ecology/health and disease and epidemiology

emerging infectious disease, community ecology, community dissimilarity, disease forecasting, pathogen biogeography

**Author for correspondence:**
Tad A. Dallas
e-mail: tad.a.dallas@gmail.com

# Testing predictability of disease outbreaks with a simple model of pathogen biogeography

Tad A. Dallas[1,2], Colin J. Carlson[3] and Timothée Poisot[4]

[1]Research Centre for Ecological Change, University of Helsinki, 00840 Helsinki, Finland
[2]Department of Biology, Louisiana State University, Baton Rouge, LA 70803, USA
[3]Department of Biology, Georgetown University, Washington, DC 20057, USA
[4]Dépt de Sciences Biologiques, Univ. de Montréal, Montréal, Canada

TAD, 0000-0003-3328-9958

Predicting disease emergence and outbreak events is a critical task for public health professionals and epidemiologists. Advances in global disease surveillance are increasingly generating datasets that are worth more than their component parts for prediction-oriented work. Here, we use a trait-free approach which leverages information on the global community of human infectious diseases to predict the biogeography of pathogens through time. Our approach takes pairwise dissimilarities between countries' pathogen communities and pathogens' geographical distributions and uses these to predict country–pathogen associations. We compare the success rates of our model for predicting pathogen outbreak, emergence and re-emergence potential as a function of time (e.g. number of years between training and prediction), pathogen type (e.g. virus) and transmission mode (e.g. vector-borne). With only these simple predictors, our model successfully predicts basic network structure up to a decade into the future. We find that while outbreak and re-emergence potential are especially well captured by our simple model, prediction of emergence events remains more elusive, and sudden global emergences like an influenza pandemic are beyond the predictive capacity of the model. However, these stochastic pandemic events are unlikely to be predictable from such coarse data. Together, our model is able to use the information on the existing country–pathogen network to predict pathogen outbreaks fairly well, suggesting the importance in considering information on co-occurring pathogens in a more global view even to estimate outbreak events in a single location or for a single pathogen.

# 1. Introduction

The emergence of infectious diseases in humans and wildlife is a continuous and natural process that is nevertheless rapidly intensifying with global change [1]. Around the world, the diversity, and frequency, of infectious outbreaks is rising over time [1,2], with an estimated 10 000 viruses with zoonotic potential that are emerging at a slow but accelerating pace [3,4]. Intensifying pathways of contact between wildlife reservoirs and humans, and the rapid spread of new pathogens among human populations around the globe are considered major drivers in this accelerating process [5,6]. Changes in climate and land use, as well as food insecurity and geopolitical conflict, are expected to exacerbate feedbacks between socio-ecological change and emerging infectious diseases (EIDs). In the face of these threats, the long-standing need to anticipate disease emergence events remains an elusive challenge for public health research [7]. Given the rise in global infectious disease outbreak events [2], understanding the factors which promote outbreak and emergence events is a primary public health concern.

The drivers of emergence events are distributed non-randomly in space and time, and follow predictable regional patterns that inherently predispose some areas to a higher burden of EIDs [8]. Different classes of emerging pathogens (e.g. new pathogens versus drug-resistant strains of familiar ones; vector-borne and/or zoonotically transmitted diseases) follow different spatial risk patterns at a global scale [1]. In part, this can be explained by the non-random distribution of host groups that disproportionately contribute to zoonotic emergence events, like bats and rodents [9,10], and are likely to continue to do so [11–13]. However, additional factors are strongly associated with the distribution of emerging infection risk; notably human population density, land cover and land use change [8], as well as a broader array of social, cultural and economic factors [14–18].

As a consequence of this heterogeneity in host distributions and other contributing factors, emerging pathogens usually follow Tobler's First Law ('near things are more related than distant things'; [19]), resulting in distinct biogeographic regions with similar host and pathogen communities [20]. However, with increasing global connectivity, both pathogens and the free-living organisms that host them are spreading around the world at an accelerating rate, and consequently, aspects of the spatial structure of pathogen diversity are becoming less pronounced. One study examining a global pathogen–country network showed that modularity is decreasing while connectance is increasing over time: pathogen ranges are on average expanding, and over time, geographically separate regions are facing more similar threats [21,22]. This biotic homogenization has critical implications for public health, as known diseases can become unfamiliar problems in novel locations, or can re-emerge in landscapes from which they were previously eradicated.

Can disease ecology be leveraged to predict these dynamics? Some recent work has examined the predictability of disease outbreak dynamics as it relates to pathogen characteristics like $R_0$ [23], but this has been studied less in spatial frameworks, and often focuses on more common outbreak events than attempting to forecast emergence events. Some forecasting approaches exist that use dozens of layers ranging from human mobility to healthcare access [24,25], but for a newly emerging disease, it may be hard to gather these mechanistic predictors, let alone anticipate which will be relevant. As an alternative method, a recent study by Murray *et al.* delineated biogeographic regions based on the popular GIDEON dataset and proposed prediction based on the principle that countries that currently share more pathogens should be more likely targets during a pathogen's future outbreaks [20]. This 'co-zone' concept shows promise as a simple way of applying ecological concepts to disease forecasting, especially if data on community structure can be built into models designed with machine learning to be explicitly predictive.

Here, we revisit the GIDEON dataset and examine the limits of predictability for this pathogen biogeography approach. We use a similarity-based approach combined with a regression approach previously used for species distribution modelling called plug-and-play, in order to forecast country -pathogen associations through time. Unlike more mechanistic approaches that simulate pathogen spread by human mobility, our model does not explicitly consider country or pathogen identity and further does not use the information on spatial structure, or trait information of countries or pathogens. Instead, the model ranks the 'suitability' of links (an unscaled measure of link likelihood) based only on the dissimilarity of pathogens' distribution and countries' pathogen community. The approach therefore lets us test the baseline predictability of the network, including how many training years are needed to make accurate predictions, and how far into the future predictions are accurate (the forecast horizon [26]).

Using our model, we also test a basic (but important) hypothesis: do outbreaks of endemic diseases have a more predictable signal than introductions and emergences (and, implicitly, are emergence events predictable)? Within emergences, we further note the subtle difference between emergence and

re-emergence and hypothesize the factors driving these might be subtly different. While both may be driven by genetic shifts in pathogens or changing land use patterns enhancing transmission risk, re-emergence events are more likely to be related to weakened healthcare infrastructure, prematurely terminated eradication campaigns [27,28] or low detection of long-term persistence of environmental pathogen reservoirs (e.g. anthrax spores in the soil and permafrost; [29,30]).

Finally, we examine whether pathogens show any differences in predictability based on agent, class or transmission mode. Diseases of zoonotic origin (i.e. with animal hosts) and with vector-borne transmission might be harder to predict due to hidden constraints on their distribution and more complicated outbreak dynamics than directly transmitted pathogens have. On the other hand, commonalities between species that share vectors or reservoir hosts might lead to similarities in distributions (a common notion in pathogen biogeography, as in how dengue models were frequently used in the early days of the Zika pandemic, given the shared vector *Aedes aegypti*; [31,32]). In this case, the community-based prediction could be more powerful for zoonotic and vector-borne diseases. Differential frequency of zoonotic and vector-borne transmission might also make different pathogen classes (viruses, bacteria, fungi and macroparasites) more or less predictable, as might different dispersal ability on a global scale, with respiratory viruses usually presumed to spread the fastest, and macroparasites generally treated as the most dispersal-limited. Understanding how the role of community structure changes for these different pathogens can help contextualize and tailor the methods we use to predict them.

# 2. Material and methods

## 2.1. Pathogen emergence data

Data from the Global Infectious Diseases and Epidemiology Network (GIDEON) contain pathogen outbreak information at the country level obtained from case reports, governmental agencies and published literature records [33,34].While there are some data for pathogen events between 1980 and 1990, the number of pathogen events reported was fewer than from 1990 onward, suggesting some potential reporting or sampling bias in these earlier years. Therefore, we restrict our analyses to yearly records between 1990 and 2016, consisting of pathogen outbreak events for 234 pathogens across 224 countries.

Records with multiple aetiological agents (e.g. 'Aeromonas and marine Vibrio infx.') and unresolved to agent level (e.g. 'Respiratory viruses—miscellaneous') were excluded from the model. In a handful of cases, we kept divisions between clinical presentations from the same pathogens, like cutaneous versus visceral leishmaniasis. Based on electronic supplementary material from [21] and updated with recent literature given several misclassifications, each was manually classified as a bacterial, viral, fungal, protozoan or macroparasitic disease, and as vector-borne and/or zoonotic or neither. In some rare cases, these were left as unknown. For example, Oropouche virus is vector-borne but its sylvatic cycle remains uncertain; similarly, the environmental origin of Bas-Congo virus is altogether unknown, despite recent speculation using machine learning with genomic data [35].

We also removed influenza from the main text analyses and place analyses containing flu in the electronic supplementary material for comparison. The most pronounced weakness of our model is that pandemic events may strongly influence model predictions, such that a pandemic of one pathogen will decrease model performance when attempting to predict the outbreak or emergence potential of other pathogens. We explore this further in the electronic supplementary material, where we see the inclusion of influenza and the corresponding 2009 flu pandemic notably limits our model performance.

## 2.2. Outbreak classes

We distinguish between three different types of pathogen events: outbreak, re-emergence and emergence. Outbreaks are recurrent pathogen events, quantified as having occurred in a given country within 3 years of a given year. Re-emergence events are those that did not occur within 3 years but have occurred at some time in a given country in the past (a cut-off we chose inspired by World Health Organization guidelines for certifying regional elimination of poliovirus or dracunculiasis). Lastly, emergence events were considered as the first record of a pathogen within a country. Given the scope of our data, a pathogen that emerged earlier than 1990, but appeared after would still be classified as an emergence event, as we only considered the 26-year period after 1990. We hypothesized that outbreaks and

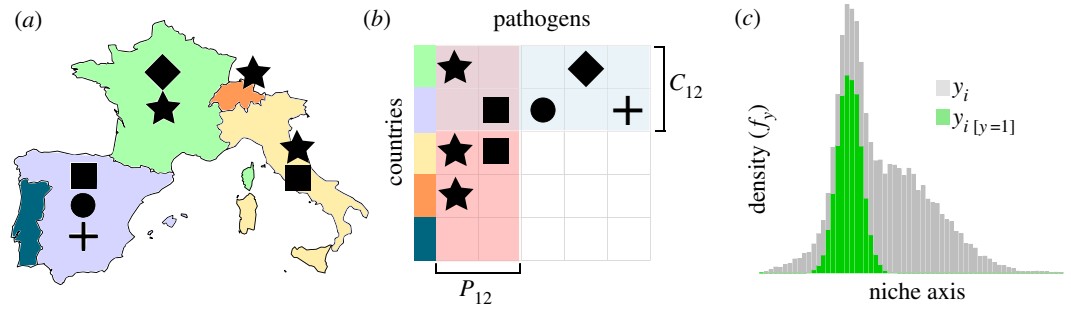

**Figure 1.** The dissimilarity-based model we used considers pathogens (indicated by symbol shape) distributed across a number of countries (*a*). Using these data, we build the interaction matrix (*b*) used to calculate pathogen community dissimilarity between two countries ($C_{ij}$) and pathogen distributional dissimilarity ($P_{ij}$). We then trained models on these two values, plus their product and the year of the sampling event. The model estimates the suitability for a pathogen to occur in a given country by comparing the density of dissimilarity values when the pathogen was present (green shaded region in *c*) relative to the dissimilarity density of all possible pairwise combinations of country and pathogen (grey density in *c*).

re-emergences should be more predictable than emergences; outbreak events occur repeatedly, providing not only ample data for model training but also a clear tendency of a pathogen to occur in a country. By contrast, emergence events are determined by many unique drivers [8], which may not be shared across any two given emergence events, and which we may need mechanistic predictors to capture.

## 2.3. Model structure

We developed a dissimilarity-based approach to forecast pathogen outbreak and emergence events that does not require detailed information on country or pathogen characteristics. Applying tools from community ecology, we calculated mean pairwise dissimilarity (Bray–Curtis index, $\overline{\mathrm{BC}}$) values for countries (how dissimilar are the pathogen communities between countries) and pathogens (how dissimilar are the geographical distributions of pathogens). For a given pair of countries *a*, *b* with $P_a$ and $P_b$ pathogens each, and *S* shared pathogens among those, the Bray–Curtis index is given as

$$\mathrm{BC}_{a,b} = 1 - \frac{2S}{P_a \times P_b}. \tag{2.1}$$

This can be treated as a measure of dissimilarity between different countries' pathogen communities. We then considered the potential for a pathogen to be found in a country proportional to the product of these dissimilarity values. We also included the year as a covariate, resulting in a set of four variables for model training; dissimilarity of the pathogen, dissimilarity of the host country, the pairwise product of these dissimilarities, and the year corresponding to those dissimilarity values.

Using these data, we applied a statistical approach previously used for species distribution modelling [36] and link prediction in ecological networks [37], referred to as the plug-and-play (PNP) approach. This approach uses information on pathogen occurrence events, and also on background interactions—country–pathogen pairs which did not have a recorded outbreak—to estimate the suitability of a country for pathogen emergence from a particular pathogen (figure 1).

We start by considering the data as a bipartite graph consisting of countries (*C*) and pathogens (*P*) that are connected by an edge when a given pathogen ($P_i$) is found in a given country ($C_j$) in a given year. This approach creates a single network for each year of outbreak monitoring. For every year and each combination of *C* and *P*, we have dissimilarity values for each country and pathogen, the product of their dissimilarities and the year of sampling (figure 1). The model functions by relating the probability density of these variables when a link is present ($f_1$) between country and pathogen to the overall density of the features for all possible country–pathogen pairs (*f*). In essence, the relationship between $f_1$ and *f* (i.e. $q = f_1/f$) represents the suitability of the country–pathogen pair given the dissimilarity and year values made conditional upon all possible country–pathogen pairs. That is, the dissimilarity and year values inform a measure akin to the potential for an outbreak event of a given pathogen to occur in a given country (i.e. the suitability for the event to occur). We estimate these suitability values (*q*) by separately estimating $f_1$ and *f* using the non-parametric kernel density estimator *npudens* from the *np* R package [38]. This method has been found to perform well in estimating missing links in host–parasite networks [39] and identifying suitable climatic areas for free-living species [36].

## 2.4. The null model

There is a possibility that a naive approach could predict pathogen outbreak events fairly well. That is, based on the frequency of pathogen occurrence within a country over time, we could estimate the probability of each country–pathogen pair by assuming it will occur with the same probability that it has occurred in the past, independent of the existing country–pathogen network. That is, if a null model were allowed to see 5 years of data, and the United States had a measles outbreak in 4 of those 5 years, a naive model would predict that a measles outbreak would occur with an 80% probability. We treated this null model as a comparison point for the accuracy metrics of our approach (light grey lines in figures).

## 2.5. Assessing model performance

We used the PNP modelling approach to address the possibility of predicting pathogen outbreak and emergence events compared to a null model. Model performance was quantified using the area under the receiver operating characteristic (AUC), a widely used metric of model accuracy that captures the ability of the classifier to rank positive instances higher than negative instances. To assess model performance, we examined three different potential scenarios.

First, we examined how the inclusion of pathogen events from previous years influenced model accuracy. That is, we predicted pathogen events of 2016 using data starting at 2015 and then including additional years until 1995. This was performed to determine the amount of data necessary to make accurate forecasts.

Second, we examined how predictive accuracy was maintained as we attempted to predict both past (hindcast) and future (forecast) pathogen events. To do this, we trained models on a 10-year period (either 2005–2015 for hindcasting, or 1990–2000 for forecasting), and used these models to predict pathogen events between 1990 and 2004 for hindcasting, and between 2001 and 2015 for forecasting.

Lastly, we examined how the accuracy of predictions might have changed over time. Given increased surveillance in more recent years, predictive accuracy might be dependent on the time period at which models are trained and predictions made. To test this, we trained models along a rolling window of 4 years from 1990–2015, using these models to predict pathogen events in the year following the final year of model training (e.g. a model trained on 1990–1994 would be used to predict pathogen events in 1995).

# 3. Results

## 3.1. Model performance

Our model performed well, with most AUC values ranging from 0.6 to 0.8 for different combinations of training and test years, pathogen subsets and emergence classes. Outbreak events were the most predictable. Across cases, we found that our dissimilarity-based model can predict outbreak events accurately, re-emergence events slightly less accurately, and emergence events only slightly better than random (i.e. AUC = 0.5). With only 5 years of data, the null model actually strongly outpredicted ours for emergences alone (figure 2) However, our approach overall outperforms the null model in all modelled scenarios, especially when temporal data were limited (figures 2–4).

Our predictive model was sensitive to the number of training years (figure 2), with accuracy plateauing around 5–10 years of training data. However, models also just trained on a single year (the temporally closest community matrix) seemed to perform disproportionately well. One possible explanation is that especially if some of the same outbreaks are spread across years and countries, the system might behave with Markov-like dynamics—that is, the present year was the primary determinant of the immediate next state.

## 3.2. Change over time

We found that the forecast horizon of the model was roughly a decade, after which outbreak predictions became no better than random (figure 3). Interestingly, predictive AUC did not decline at the same rate when hindcasting and forecasting. Model accuracy was higher when hindcasting over a comparable temporal horizon, with accurate predictions up to 15 years in the past (figure 3). This perhaps indicates that as the country–pathogen network becomes asymptotically more connected and stable [22], the network accumulates information content, reducing the time sensitivity of hindcasting performance.

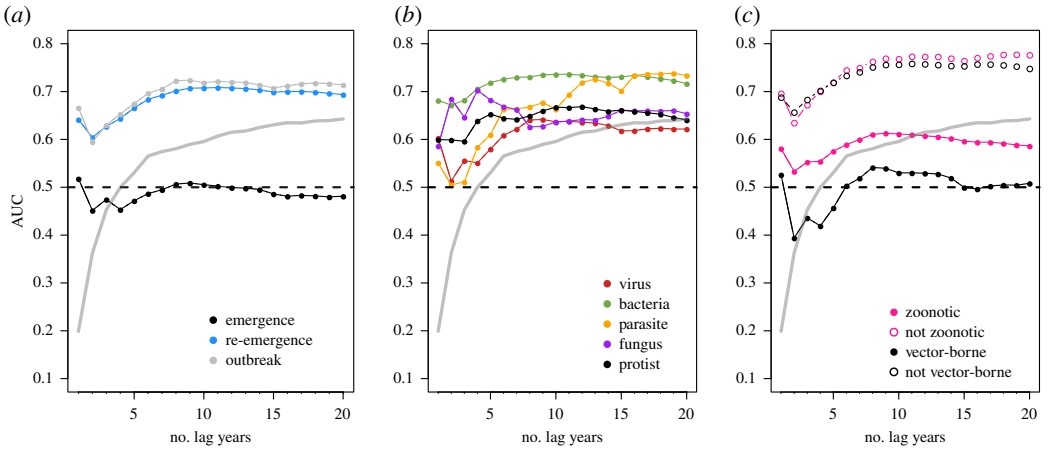

**Figure 2.** Pathogen events from previous years increased model predictive accuracy after an initial small decrease, suggesting that 5 years or more of data improves predictions, but accuracy could actually decrease in some data-sparse situations where only 2 or 3 years of data were available. Performance of the null expectation (grey line) was less than our approach, except when the null was given more than 15 years of previous data.

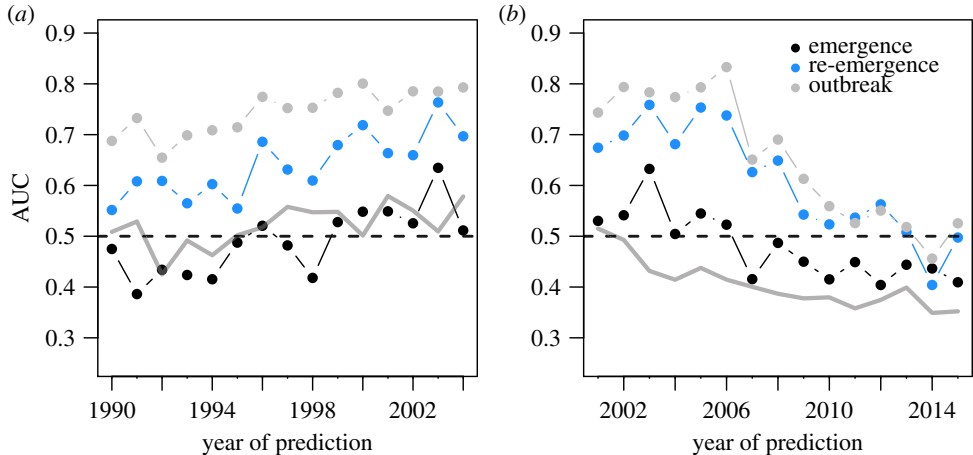

**Figure 3.** Predictive accuracy decreased when attempting to forecast far into the past or future. Models were trained on either the period between 2005 and 2015 (for prediction into the past) or 1990 and 2000 (for prediction into the future). The null expectation (grey line) performed consistently worse than our approach.

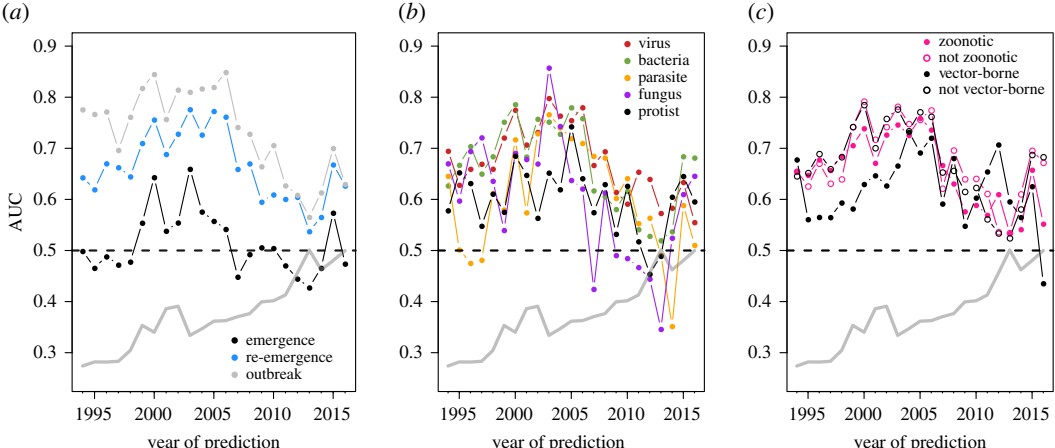

**Figure 4.** Using a rolling window ($t = 4$ years), we found that predictive accuracy did not increase as a result of enhanced surveillance and data collection of more recent years. The null expectation (grey line) performed consistently worse than our approach.

Examining a rolling window of $t$ years ($t = 4$ years) over the last two decades, we failed to detect evidence that the enhanced reporting and surveillance in more recent years improved our model's predictive ability (figure 4). This also suggests that even though there were annual variations in the sample size of both pathogens and countries, there was still consistency in the structure of the country–pathogen interaction matrix over time. We explore the sensitivity of this finding to the size of the rolling window in the electronic supplementary material.

## 3.3. Pathogen differences in predictability

Differences in PNP model accuracy among pathogen types existed when examining the effect of the amount of data used for model training (figure 2), with viruses having lower accuracy relative to bacteria, fungi or other parasites. The simplest explanation for this is that accuracy is sensitive to the number of events. However, the average number of viral occurrences over time ($\bar{x} = 179$) was only slightly less than the average number of bacterial ($\bar{x} = 185$) occurrences, and far greater than the average number of fungal ($\bar{x} = 10$), macroparasite ($\bar{x} = 17.7$) or protozoan parasite ($\bar{x} = 22.5$) occurrence events. The average number of pathogen occurrences over time is qualitatively proportional to the number of unique viruses ($n = 83$), bacteria ($n = 81$), fungi ($n = 14$), macroparasites ($n = 38$) and protozoans ($n = 15$) we examined. Interestingly, differences among pathogen types were not found when examining the ability of the modelling approach to hindcast/forecast (figure 3) or when examining predictive accuracy along a rolling window (figure 4).

For our 2016 explanatory PNP model, differentiating pathogens based on zoonotic and vector-borne transmission modes suggested that both classes of pathogens were more difficult to forecast (figure 2). While it is possible that class imbalances between groups might drive this pattern (i.e. more event occurrences may increase model predictive accuracy), this seems unlikely: the majority of pathogens (144 of 228) were zoonotic, and many (59 of 233) were vector-borne. A more compelling explanation is that this year was an anomalous result; transmission mode did not influence accuracy when hindcasting/forecasting (electronic supplementary material, figure S5) or when models were trained along a rolling window (figure 4), though there was notable year-to-year variation in the latter.

# 4. Discussion

Community ecology and biogeography have a history as deeply linked fields and both play an increasingly significant role in emerging infectious disease research. [20,40,41] However, both of these fields have only recently begun to consider pathogen communities explicitly [41]. Our goal was to examine whether the intrinsic structure of pathogen biogeography, approached as a bipartite network, was predictable enough to enable forecasting of different outbreak types—even in the absence of any other mechanistic predictors, like transmission mode, phylogenetic data, environmental covariates or even the identities of the countries and pathogens themselves. That is, based solely on the dissimilarity of pathogens in their distributions and of countries in their pathogen communities, can we successfully predict outbreak, re-emergence and emergence events?

Despite obvious stochasticity and data limitations, the modelling approach performed well with as little as 7 to 10 years of training data, and when predicting country–pathogen network structure across large time windows. The model was able to capture pathogen outbreak and re-emergence potential well, suggesting that, at least at broad spatial scales, pathogen outbreak and re-emergence events are both recurrent and predictable (and that community assembly is structured and predictive of outbreak potential). However, our model generally failed to forecast pathogen emergence events. This is largely unsurprising, as predicting when and where the next major public health threat will emerge is an incredibly difficult task which remains unsolved despite having received decades of attention [1,7,8]. However, the failure of community information to anticipate local emergences is still disappointing, especially given that biogeographic 'co-zones' could be useful strategic tools in the absence of knowledge about a newly described pathogen [17]. It remains possible that other formulations might be more successful (e.g. restricting the pathogen–country matrix to a particular subset of closely related pathogens) or that community structure might improve model performance alongside more mechanistic predictors.

We found some indications of differences in the predictability of pathogen events as a function of pathogen type and transmission modes. In the 2016 model breakdown, bacteria were the most predictable while viruses were disproportionately unpredictable, as were zoonotic and vector-borne pathogens. Given how clearly unpredictable emergence events were, this might make intuitive sense:

zoonotic pathogens make up the majority of emerging diseases [1], and single-stranded RNA viruses (many vector-borne) have been responsible for many of the biggest recent emergence events [9]. However, this pattern did not appear to hold up across all or even most years, and the factors that reduce model performance on a year-by-year basis are mostly unclear at the community level.

One contributor to interannual variation is large-scale events such as pandemics, which appeared to strongly influence the prediction of the entire country–pathogen network. While pandemic spread may be predictable using detailed information on climate, human movement and local environmental suitability [7,42,43], our approach lacks these mechanistic predictors and is sensitive to these black swan events. This can be seen in reduced model performance during the 2009 flu pandemic, including for pathogens with no relationship to flu, although viruses and vector-borne pathogens are more severely affected (electronic supplementary material). So while the model benefits from pathogen community data, rare and widespread events can strongly reduce model accuracy. Future work to differentially weight these stochastic events would probably improve model performance.

While this approach improves the estimation of outbreak and emergence potential relative to the null model—particularly for rare pathogens or poorly sampled countries—it is also worth noting that our approach is *not* a valid stand-alone forecasting tool. This is in large part due to how time is used in the model: though year is a covariate, the model itself is not temporally explicit, meaning that the model can predict a certain link following on previous years, but it would be erroneous to interpret that as a forecast for a given point in time. However, the tool can be used to investigate pathogen outbreak and emergence potential under different pathogen range expansion scenarios. That is, by simulating pathogen spread into new geographical locations, the model could be used to identify potential pathogens that may also shift their geographical ranges. Since the method is based on dissimilarity of countries and pathogen distributions, it is possible to examine the expected outcome as pathogen distributions become more (or less) homogeneous, or countries become more (or less) dissimilar in their pathogen communities. Finally, the sensitivity of the model to pathogen shifts, transients or imperfect detection are all things that can be tested using simulation approaches and may provide further insight into our ability to forecast outbreak and emergence events.

Within infectious disease ecology, a disproportionate focus has emerged on the drivers and predictability of emergence events [8]. Recent work offers a compelling case that community ecology might bring predictive tools to bear on that problem [40], and modelling work suggests that community assembly data can be leveraged to better predict how pathogens spread [20], the host range of emerging diseases [9,44] and the dynamics of diseases within an ecosystem [45,46]. Our results show how a simple model considering the entire pathogen community captures important global geographical variation in outbreak potential, but as a stand-alone tool, still struggles to predict when a pathogen will first arrive in a new region. For this definition of emergence (and this dataset), we suggest this indicates an outer bound of predictive capacity based *only* on community structure; however, community data could be very easily leveraged alongside other socio-ecological predictors to forecast disease emergence as an ecosystem process rather than a single-species one. Finally, this simple model provides a basic benchmark for more complex models which may require more detailed data, which is not readily available.

Ethics. This study used existing data on pathogen outbreak and emergence events in human populations.

Data accessibility. R code is available on figshare at https://doi.org/10.6084/m9.figshare.6364955.

Authors' contributions. T.A.D., C.J.C. and T.P. conceived of the idea for the study. T.A.D. and T.P. contributed to model development. All authors contributed to the writing of the manuscript.

Competing interests. The authors declare no competing interests.

Funding. This work was supported by the National Socio-Environmental Synthesis Center (SESYNC) under funding received from the National Science Foundation (grant no. DBI-1639145), and by a Georgetown Environment Initiative postdoctoral fellowship.

Acknowledgements. We thank GIDEON (https://www.gideononline.com/) for their collection and curation of the data, and the Bansal Lab at Georgetown for the thoughtful feedback.

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
