## [Reviewer comments · Royal Society Open Science]

Review History

RSOS-190883.R0 (Original submission)

Review form: Reviewer 1

Is the manuscript scientifically sound in its present form?

Yes

Are the interpretations and conclusions justified by the results?

No

Is the language acceptable?

No

Is it clear how to access all supporting data?

Yes

Do you have any ethical concerns with this paper?

No

Have you any concerns about statistical analyses in this paper?

No

Recommendation?

Major revision is needed (please make suggestions in comments)

Comments to the Author(s)

I was pleased to see the revisions made by the authors in response to the first round of reviews. However, I would like them to go further in order to fully address the two main issues previously raised: this is essentially a black box applied to a large dataset (of which nothing is shown), and the outcome is completely opaque.

Regarding the methods, the authors have greatly improved their description, but it's still a verbal summary that leaves a lot of details out. Frustratingly, there is only one equation for the Bray-Curtis index for one pair of countries, and all the next steps are just mentioned. Granted, the authors have shared a very neat Rmd file allowing anyone to rerun their code, but that's still fairly opaque. What I would really like is a complete description of the methods with the equations in the Supplementary Materials that will tell me how the mean dissimilarity values were calculated and how everything else follows.

The Results give a detailed account of the model's performance based on AUC, but, as I've said before, we are none the wiser regarding what can be learned from this model. In my experience, most models can be used for one of two purposes: either for inference from a dataset or for prediction/forecast. There is no inference presented based on the analysis of the existing dataset, and we are then warned that, despite the model's good performance, it should not be used for forecast. So what's the point? It's a bit like applying a GLM to quantify risk factors and only reporting the overall model performance with an R² and a p-value, hiding the data and the odds ratios and then warning the readers that, even though it's a good model and they can download the code, they shouldn't use it for predictive purposes. I'm sorry but I still don't get it.

Review form: Reviewer 2 (Kris Murray)

Is the manuscript scientifically sound in its present form?

Yes

Are the interpretations and conclusions justified by the results?

Yes

Is the language acceptable?

Yes

Is it clear how to access all supporting data?

Yes

Do you have any ethical concerns with this paper?

No

Have you any concerns about statistical analyses in this paper?

No

Recommendation?

Accept with minor revision (please list in comments)

Comments to the Author(s)

Review of Dallas et al.,

General comments

I found this ms to be interesting, well written, relevant and in general a nice inter-disciplinary study and a solid addition to the current literature on the macroecology/biogeography of human infectious diseases. Although not a reviewer previously, I was able to read the previous reviews from Proc B and feel that the majority of the concerns raised there have been adequately addressed in this revision. Indeed, I might go so far as to suggest that the decision from Proc B to deny resubmission and pass the ms instead to their Open Science sister journal following major revisions was perhaps too hasty in this case, given revisions were mostly focussed on improving description. The ms is vibrant and brimming with nice ideas and insights and, importantly, actually tests some predictions about disease emergence that can be evaluated and validated, where many other studies on this topic simply reinforce self-perpetuating ideas.

Specific comments

Personally, I was pleased to see the study make reference and some explicit tests relevant to a general assertion put forward in Murray et al., (2015; PNAS) regarding the potential to use co-occurrence (“co-zones”) / beta-diversity / dissimilarity (and biogeographic approaches more generally) for risk assessment, surveillance or even, as tested here, prediction for ‘emerging’ human infections. However, at the distinct risk of appearing self-interested (sorry!), I felt the Introduction / Methods didn’t really do justice to framing the clear parallels between the studies in terms of the questions being tested (potential utility for risk assessment / prediction of outbreaks or emergence from disease assemblage/community data), the methods (agnostic, trait-free dissimilarity / co-occurrence metrics) and the data (GIDEON subset based on similar exclusion criteria). The Discussion was more explicit (albeit somewhat caricatured – see comment below) in referencing e.g., “co-zones”. There are very few studies taking this sort of an approach for human infectious diseases, so it seems the similarities to existing literature could be more plainly stated up front, such that the novelty of the contribution can be more clearly emphasised.

P6 L99 “our model does not use information on pathogen identity, country identity, pathogen traits, or spatial structure. Instead, the model ranks the “suitability” of links (an unscaled measure of link likelihood) based only on the dissimilarity of pathogens’ distribution[s] and countries’ pathogen community.” – This is confusing – you need pathogen and country identities to calculate dissimilarity, so I don’t follow how the model doesn’t use these.

P9 L26 – “... that does not require country-level or pathogen traits data” – a bit confusing since the outbreak database in GIDEON is country-level, no?

P15 L333 – “However, the failure of community information to help anticipate local emergences is still disappointing, especially given the proposal that biogeographic “co-zones” could be useful strategic tools for pandemic forecasting (17).” – see my general comment above about this. Also the ref here is incorrect if you are referring to the “co-zones” proposed by Murray et al., 2015 PNAS. Finally, that paper does not talk about applying co-zones to pandemic forecasting specifically, and your analysis does not focus on pandemic forecasting either so sentence seems a bit misplaced.

In any case, I don't feel that the analysis rules this general idea out as, as broadly acknowledged, there are other methodological formulations of the general approach, different data sets other than GIDEON that could be used or strived for, and additional predictions arising from the hypotheses that remain to be tested. In this respect, I find the tests a bit too narrow for a conclusion that seems a bit too definitive. I would perhaps suggest being more cautious and stick to exactly what has been examined.

For example, even the definition of 'emergence' being employed and tested here could be pressed further. Here it is the 'first report in a country', which I'm fine with as it is at least specific. However, 'emergence' has also been considered as the novel appearance of a disease (anywhere), increasing host range, increasing geographic distribution (invasion, long-distance jumps), increasing incidence and so on, and in many cases all of these things all munged together. In this context, I think there is more scope here to be even more specific about what elements of 'emergence' can and can't be predicted with this approach, and then of course tempering the conclusions to match.

P16 L365 - "That is, researchers could construct artificial data which differs from empirical data slightly, and quantify the ability of the model to predict those novel events." - I didn't really follow this bit, suggest further clarification.

P17 L380 - "Though this casts doubt on biogeographic tools like "co-zones" as standalone tools for surveillance or outbreak response, our study is a compelling indicator that community data could be very easily leveraged alongside other socioecological predictors to forecast disease emergence as an ecosystem process rather than a single-species one." - No ref for "co-zones" but as above, if you are referring to Murray et al., 2015 I don't think I implied that this should be a stand-alone tool but rather an approach that could provide some of the earliest insights or prior expectations in data poor situations for surveillance or outbreak response.

Your study provides further evidence to support this, at least for outbreaks and re-emergence, but also highlights something of a prediction ceiling so to speak for local emergences and long time horizons. So again this sentence seems a bit misplaced or caricatured. It rather appears we make the same basic points about the potential to leverage community data more effectively.

Also, I think there could be a useful distinction with these approaches between the 'where' and 'when' of prediction that could be better drawn out. Historical co-occurrence has clear predictive value for where things might turn up in future based on where they have turned up in the past but a much tougher question, as also touched upon here, is when it will turn up.

I hope some of these thoughts will be useful for your revisions.
Kris Murray

Decision letter (RSOS-190883.R0)

10-Jul-2019

Dear Dr Dallas,

The editors assigned to your paper ("Testing predictability of disease outbreaks with a simple model of pathogen biogeography") have now received comments from reviewers. We would like you to revise your paper in accordance with the referee and Associate Editor suggestions which

can be found below (not including confidential reports to the Editor). Please note this decision does not guarantee eventual acceptance.

Please submit a copy of your revised paper before 02-Aug-2019. Please note that the revision deadline will expire at 00.00am on this date. If we do not hear from you within this time then it will be assumed that the paper has been withdrawn. In exceptional circumstances, extensions may be possible if agreed with the Editorial Office in advance. We do not allow multiple rounds of revision so we urge you to make every effort to fully address all of the comments at this stage. If deemed necessary by the Editors, your manuscript will be sent back to one or more of the original reviewers for assessment. If the original reviewers are not available, we may invite new reviewers.

- Data accessibility

<http://datadryad.org/submit?journalID=RSOS&manu=RSOS-190883>

- Competing interests

- Authors' contributions

- Acknowledgements

- Funding statement

on behalf of Kevin Padian (Subject Editor)
openscience@royalsociety.org

Editor's comments to Author:

Thanks for your submission. As you will see both reviewers are happy with the revisions you've made, and both have further concerns that we'd like to give you some time to address. Please respond to these specifically in your revision, and best wishes.

Comments to Author:

Reviewers' Comments to Author:

Reviewer: 1

Comments to the Author(s)

I was pleased to see the revisions made by the authors in response to the first round of reviews. However, I would like them to go further in order to fully address the two main issues

previously raised: this is essentially a black box applied to a large dataset (of which nothing is shown), and the outcome is completely opaque.

Regarding the methods, the authors have greatly improved their description, but it's still a verbal summary that leaves a lot of details out. Frustratingly, there is only one equation for the Bray-Curtis index for one pair of countries, and all the next steps are just mentioned. Granted, the authors have shared a very neat Rmd file allowing anyone to rerun their code, but that's still fairly opaque. What I would really like is a complete description of the methods with the equations in the Supplementary Materials that will tell me how the mean dissimilarity values were calculated and how everything else follows.

The Results give a detailed account of the model's performance based on AUC, but, as I've said before, we are none the wiser regarding what can be learned from this model. In my experience, most models can be used for one of two purposes: either for inference from a dataset or for prediction/forecast. There is no inference presented based on the analysis of the existing dataset, and we are then warned that, despite the model's good performance, it should not be used for forecast. So what's the point? It's a bit like applying a GLM to quantify risk factors and only reporting the overall model performance with an R² and a p-value, hiding the data and the odds ratios and then warning the readers that, even though it's a good model and they can download the code, they shouldn't use it for predictive purposes. I'm sorry but I still don't get it.

Reviewer: 2

Comments to the Author(s)
Review of Dallas et al.,

General comments

I found this ms to be interesting, well written, relevant and in general a nice inter-disciplinary study and a solid addition to the current literature on the macroecology/biogeography of human infectious diseases. Although not a reviewer previously, I was able to read the previous reviews from Proc B and feel that the majority of the concerns raised there have been adequately addressed in this revision. Indeed, I might go so far as to suggest that the decision from Proc B to deny resubmission and pass the ms instead to their Open Science sister journal following major revisions was perhaps too hasty in this case, given revisions were mostly focussed on improving description. The ms is vibrant and brimming with nice ideas and insights and, importantly, actually tests some predictions about disease emergence that can be evaluated and validated, where many other studies on this topic simply reinforce self-perpetuating ideas.

Specific comments

Personally, I was pleased to see the study make reference and some explicit tests relevant to a general assertion put forward in Murray et al., (2015; PNAS) regarding the potential to use co-occurrence ("co-zones") / beta-diversity / dissimilarity (and biogeographic approaches more generally) for risk assessment, surveillance or even, as tested here, prediction for 'emerging' human infections. However, at the distinct risk of appearing self-interested (sorry!), I felt the Introduction / Methods didn't really do justice to framing the clear parallels between the studies in terms of the questions being tested (potential utility for risk assessment / prediction of outbreaks or emergence from disease assemblage/community data), the methods (agnostic, trait-free dissimilarity / co-occurrence metrics) and the data (GIDEON subset based on similar exclusion criteria). The Discussion was more explicit (albeit somewhat caricatured – see comment below) in referencing e.g., "co-zones". There are very few studies taking this sort of an approach

for human infectious diseases, so it seems the similarities to existing literature could be more plainly stated up front, such that the novelty of the contribution can be more clearly emphasised.

P6 L99 “our model does not use information on pathogen identity, country identity, pathogen traits, or spatial structure. Instead, the model ranks the “suitability” of links (an unscaled measure of link likelihood) based only on the dissimilarity of pathogens’ distribution[s] and countries’ pathogen community.” – This is confusing – you need pathogen and country identities to calculate dissimilarity, so I don’t follow how the model doesn’t use these.

P9 L26 – “... that does not require country-level or pathogen traits data” – a bit confusing since the outbreak database in GIDEON is country-level, no?

P15 L333 – “However, the failure of community information to help anticipate local emergences is still disappointing, especially given the proposal that biogeographic “co-zones” could be useful strategic tools for pandemic forecasting (17).” – see my general comment above about this. Also the ref here is incorrect if you are referring to the “co-zones” proposed by Murray et al., 2015 PNAS. Finally, that paper does not talk about applying co-zones to pandemic forecasting specifically, and your analysis does not focus on pandemic forecasting either so sentence seems a bit misplaced.

In any case, I don’t feel that the analysis rules this general idea out as, as broadly acknowledged, there are other methodological formulations of the general approach, different data sets other than GIDEON that could be used or strived for, and additional predictions arising from the hypotheses that remain to be tested. In this respect, I find the tests a bit too narrow for a conclusion that seems a bit too definitive. I would perhaps suggest being more cautious and stick to exactly what has been examined.

For example, even the definition of ‘emergence’ being employed and tested here could be pressed further. Here it is the ‘first report in a country’, which I’m fine with as it is at least specific. However, ‘emergence’ has also been considered as the novel appearance of a disease (anywhere), increasing host range, increasing geographic distribution (invasion, long-distance jumps), increasing incidence and so on, and in many cases all of these things all munged together. In this context, I think there is more scope here to be even more specific about what elements of ‘emergence’ can and can’t be predicted with this approach, and then of course tempering the conclusions to match.

P16 L365 – “That is, researchers could construct artificial data which differs from empirical data slightly, and quantify the ability of the model to predict those novel events.” – I didn’t really follow this bit, suggest further clarification.

P17 L380 – “Though this casts doubt on biogeographic tools like “co-zones” as standalone tools for surveillance or outbreak response, our study is a compelling indicator that community data could be very easily leveraged alongside other socioecological predictors to forecast disease emergence as an ecosystem process rather than a single-species one.” – No ref for “co-zones” but as above, if you are referring to Murray et al., 2015 I don’t think I implied that this should be a stand-alone tool but rather an approach that could provide some of the earliest insights or prior expectations in data poor situations for surveillance or outbreak response.

Your study provides further evidence to support this, at least for outbreaks and re-emergence, but also highlights something of a prediction ceiling so to speak for local emergences and long time horizons. So again this sentence seems a bit misplaced or caricatured. It rather appears we make the same basic points about the potential to leverage community data more effectively.

Also, I think there could be a useful distinction with these approaches between the 'where' and 'when' of prediction that could be better drawn out. Historical co-occurrence has clear predictive value for where things might turn up in future based on where they have turned up in the past but a much tougher question, as also touched upon here, is when it will turn up.

I hope some of these thoughts will be useful for your revisions.
Kris Murray

Author's Response to Decision Letter for (RSOS-190883.R0)

See Appendix A.

RSOS-190883.R1 (Revision)

Review form: Reviewer 1

Is the manuscript scientifically sound in its present form?

Yes

Are the interpretations and conclusions justified by the results?

Yes

Is the language acceptable?

Yes

Do you have any ethical concerns with this paper?

No

Have you any concerns about statistical analyses in this paper?

No

Recommendation?

Accept as is

Comments to the Author(s)

I would like to thank the authors for the detailed rebuttal and the helpful clarifications they made in the manuscript. I now have a better understanding and appreciation of the workings and usefulness of the model, particularly with reference to the paper by Murray et al which I wasn't familiar with. I have no reservation in recommending that the manuscript should be published in its current form.

Review form: Reviewer 2 (Kris Murray)

Is the manuscript scientifically sound in its present form?

Yes

Are the interpretations and conclusions justified by the results?

Yes

Is the language acceptable?

Yes

Do you have any ethical concerns with this paper?

No

Have you any concerns about statistical analyses in this paper?

No

Recommendation?

Accept as is

Comments to the Author(s)

The authors have made a very solid revision and addressed all the points previously raised.

Decision letter (RSOS-190883.R1)

08-Oct-2019

Dear Dr Dallas,

I am pleased to inform you that your manuscript entitled "Testing predictability of disease outbreaks with a simple model of pathogen biogeography" is now accepted for publication in Royal Society Open Science.

Kind regards,

Lianne Parkhouse
Royal Society Open Science
openscience@royalsociety.org

on behalf of the Associate Editor, and Professor Kevin Padian (Subject Editor)
openscience@royalsociety.org

Reviewer comments to Author:

Reviewer: 1
Comments to the Author(s)

I would like to thank the authors for the detailed rebuttal and the helpful clarifications they made in the manuscript. I now have a better understanding and appreciation of the workings and usefulness of the model, particularly with reference to the paper by Murray et al which I wasn't familiar with. I have no reservation in recommending that the manuscript should be published in its current form.

Reviewer: 2
Comments to the Author(s)

The authors have made a very solid revision and addressed all the points previously raised.

Appendix A

Thanks for your submission. As you will see both reviewers are happy with the revisions you've made, and both have further concerns that we'd like to give you some time to address. Please respond to these specifically in your revision, and best wishes.

Thank you for your efforts on this manuscript. We have attempted to address the reviewer concerns, some of which we admittedly push back against. Please see the following responses for more detailed responses to reviewer concerns.

Reviewer: 1

I was pleased to see the revisions made by the authors in response to the first round of reviews. However, I would like them to go further in order to fully address the two main issues previously raised: this is essentially a black box applied to a large dataset (of which nothing is shown), and the outcome is completely opaque.

We are happy that the reviewer is pleased with our corrections. We feel that the two remaining issues are not quite as important as the reviewer makes them sound, and will briefly outline our reasoning here (as well as indicate which steps we took to clarify the main text). The model is not a black box, and an expanded paragraph was added to describe the model structure in the main text, as well as the inclusion of several citations of previous work using the approach. To attempt to address this first issue, we have added supplemental description of our analytical approach, as the reviewer suggests below. To address the second point, we have edited the text in several places to try to highlight the utility of the approach as well as the numerous testable hypotheses generated from our approach; as such, we do not think it is fair to call the outcome "opaque".

Regarding the methods, the authors have greatly improved their description, but it's still a verbal summary that leaves a lot of details out. Frustratingly, there is only one equation for the Bray-Curtis index for one pair of countries, and all the next steps are just mentioned. Granted, the authors have shared a very neat Rmd file allowing anyone to rerun their code, but that's still fairly opaque. What I would really like is a complete description of the methods with the equations in the Supplementary Materials that will tell me how the mean dissimilarity values were calculated and how everything else follows.

In our previous edits, we tried to discuss the "plug-and-play" model in more detail, while presenting it in a more narrative manner; in particular, this is because the model was already explained in great detail in a previous publication. That being said, the paragraph starting on line 201 has a fair amount of inline equations with clear verbal descriptions of how the model is structured. We also cite the relevant paper first pushing forward the "plug-and-play" approach for species distribution modeling, and for link prediction in ecological networks. We have now added some supplemental material describing the data structure and the modeling approach in more detail. However, we believe the main text description of the model is sufficient for readers to understand the approach, and, as the reviewer notes, the code is well documented and freely available.

The Results give a detailed account of the model's performance based on AUC, but, as I've said before, we are none the wiser regarding what can be learned from this model. In my experience, most models can be used for one of two purposes: either for inference from a dataset or for prediction/forecast. There is no inference presented based on the analysis of the existing dataset, and we are then warned that, despite the model's good performance, it should not be used for forecast. So what's the point? It's a bit like applying a GLM to quantify risk factors and only reporting the overall model performance with an R² and a p-value, hiding the data and the odds ratios and then warning the readers that, even though it's a good model and they can download the code, they shouldn't use it for predictive purposes. I'm sorry but I still don't get it.

The caveat about prediction is because the model requires information on the presence of other country-pathogen pairs, so it's useful in a "what-if" scenario, for hindcasting, or for a variety of other reasons; all of these tasks fall under the general umbrella of "prediction", but we strongly feel that it would be intellectually disingenuous not to add a clarification about limitations. The reviewer suggests that models can be used for inference or prediction (akin to Brieman's "two cultures"). We suggest that inference is gained from our model, which we attempt to

emphasize throughout. Using a novel approach to estimate the suitability of country-pathogen pairs in the absence of trait information, we can use model performance as a function of pathogen characteristics (Figure 2) or time (Figures 3 and 4). First, the ability to capture country-pathogen suitability values without any traits is intriguing. Second, the forecast horizon (how far into the future or back into the past can we predict country-pathogen network structure at the global scale?) is largely unknown for infectious disease outbreaks (i.e., how much information do we need to estimate the suitability of a country for an outbreak or emergence event of a particular pathogen?). We have tried to emphasize this in the revised manuscript.

We apologize that it seems opaque, but we have done our best to present the material in a clear way.

Reviewer: 2

Review of Dallas et al.,

General comments

I found this ms to be interesting, well written, relevant and in general a nice inter-disciplinary study and a solid addition to the current literature on the macroecology/biogeography of human infectious diseases. Although not a reviewer previously, I was able to read the previous reviews from Proc B and feel that the majority of the concerns raised there have been adequately addressed in this revision. Indeed, I might go so far as to suggest that the decision from Proc B to deny resubmission and pass the ms instead to their Open Science sister journal following major revisions was perhaps too hasty in this case, given revisions were mostly focussed on improving description. The ms is vibrant and brimming with nice ideas and insights and, importantly, actually tests some predictions about disease emergence that can be evaluated and validated, where many other studies on this topic simply reinforce self-perpetuating ideas.

We thank the reviewer for their thoughtful comments, and their kind words.

Specific comments

Personally, I was pleased to see the study make reference and some explicit tests relevant to a general assertion put forward in Murray et al., (2015; PNAS) regarding the potential to use co-occurrence (“co-zones”) / beta-diversity / dissimilarity (and biogeographic approaches more generally) for risk assessment, surveillance or even, as tested here, prediction for ‘emerging’ human infections. However, at the distinct risk of appearing self-interested (sorry!), I felt the Introduction / Methods didn’t really do justice to framing the clear parallels between the studies in terms of the questions being tested (potential utility for risk assessment / prediction of outbreaks or emergence from disease assemblage/community data), the methods (agnostic, trait-free dissimilarity / co-occurrence metrics) and the data (GIDEON subset based on similar exclusion criteria). The Discussion was more explicit (albeit somewhat caricatured – see comment below) in referencing e.g., “co-zones”. There are very few studies taking this sort of an approach for human infectious diseases, so it seems the similarities to existing literature could be more plainly stated up front, such that the novelty of the contribution can be more clearly emphasised.

We agreed the parallels should have been made clearer, given the small number of modeling studies with this approach/scope. We have restructured a paragraph of the Introduction to make the parallels clearer, and set up the idea that our paper examines the limits of predictability for the framework Murray proposed:

“Can disease ecology be leveraged to predict these dynamics? Some recent work has examined the predictability of disease outbreak dynamics as it relates to pathogen characteristics like R_0 {scarpino2019predictability}, but this has been studied less in spatial frameworks, and even less in terms of the long-term process of emergence. Some forecasting approaches exist that utilize dozens of layers ranging from human mobility to healthcare access {pigott2017local,khan2009spread}, but for a newly-emerging disease, it may be hard to gather these mechanistic predictors, let alone anticipate which will be relevant. As an alternative method, a recent study by Murray *et al.*

delineated biogeographic regions based on the popular GIDEON dataset, and proposed prediction based on the principle that countries that currently share more pathogens should be more likely targets during a pathogen’s future outbreaks. {murray2015global} This “co-zone” concept shows promise as a simple way of applying ecological concepts to disease forecasting, especially if data on community structure can be built into models designed with machine learning to be explicitly predictive. Here, we revisit the GIDEON dataset, and examine the limits of predictability for this pathogen biogeography approach. . . .”

P6 L99 “our model does not use information on pathogen identity, country identity, pathogen traits, or spatial structure. Instead, the model ranks the “suitability” of links (an unscaled measure of link likelihood) based only on the dissimilarity of pathogens’ distribution[s] and countries’ pathogen community.” – This is confusing – you need pathogen and country identities to calculate dissimilarity, so I don’t follow how the model doesn’t use these.

This was an overstatement. While pathogen and country identity aren’t explicitly incorporated into the model, the reviewer is correct that a country or pathogen will have a (potentially not unique) value of mean dissimilarity to other countries or pathogens. We have edited the sentence to read:

“Unlike more mechanistic approaches that simulate pathogen spread by human mobility, our model does not explicitly consider country or pathogen identity, and further does not use information on spatial structure, or trait information of countries or pathogens.”

P9 L26 – “. . . that does not require country-level or pathogen traits data” – a bit confusing since the outbreak database in GIDEON is country-level, no?

This is a similar misstep as above, where we wanted to highlight the trait-free nature of the model, but it sounds confusing. We have edited this sentence to read:

“We developed a dissimilarity-based approach to forecast pathogen outbreak and emergence events that does not require detailed information on country or pathogen characteristics.”

P15 L333 – “However, the failure of community information to help anticipate local emergences is still disappointing, especially given the proposal that biogeographic “co-zones” could be useful strategic tools for pandemic forecasting (17).” – see my general comment above about this. Also the ref here is incorrect if you are referring to the “co-zones” proposed by Murray et al., 2015 PNAS. Finally, that paper does not talk about applying co-zones to pandemic forecasting specifically, and your analysis does not focus on pandemic forecasting either so sentence seems a bit misplaced.

In any case, I don’t feel that the analysis rules this general idea out as, as broadly acknowledged, there are other methodological formulations of the general approach, different data sets other than GIDEON that could be used or strived for, and additional predictions arising from the hypotheses that remain to be tested. In this respect, I find the tests a bit too narrow for a conclusion that seems a bit too definitive. I would perhaps suggest being more cautious and stick to exactly what has been examined.

This is true, and honestly is one of the big benefits that we see from the model approach, in that it’s flexible enough to be run on most datasets, and has been used previously as a species distribution model (Drake and Richards 2017) and a host-parasite link prediction tool (Dallas et al. 2017). We have softened the wording with respect to the co-zone sections (specifically around line 333), and better represented the intention of the Murray et al. (2015) paper. We also now qualify our model’s failure with the broader possibility that other formulations might be more successful.

Our statement now reads “However, the failure of community information to anticipate local emergences is still disappointing, especially given that biogeographic “co-zones” could be useful strategic tools in the absence of knowledge about a newly-described pathogen {murray2010historical}. It remains possible that other formulations might be more successful (e.g., restricting the pathogen-country matrix to a particular subset of closely-related pathogens), or that community structure might improve model performance alongside more mechanistic predictors.”

For example, even the definition of ‘emergence’ being employed and tested here could be pressed further. Here it is the ‘first report in a country’, which I’m fine with as it is at least specific. However, ‘emergence’ has also been considered as the novel appearance of a disease (anywhere), increasing host range, increasing geographic distribution (invasion, long-distance jumps), increasing incidence and so on, and in many cases all of these things all munged together. In this context, I think there is more scope here to be even more specific about what elements of ‘emergence’ can and can’t be predicted with this approach, and then of course tempering the conclusions to match.

We agree that there are several ways emergence can be conceptualized, all important, and the challenge of telling them apart frequently makes it hard to pull apart what ecologists are talking about in recent literature. This is why we make a deliberate effort to clearly operationalize “emergence” as a data structure that is compatible with the structure of models we are considering, and introduce this idea in the Introduction and Methods. Introducing and testing some of these other definitions would probably make it harder to understand the limits of our model, and therefore, we considered it beyond the scope of this manuscript.

P16 L365 – “That is, researchers could construct artificial data which differs from empirical data slightly, and quantify the ability of the model to predict those novel events.” – I didn’t really follow this bit, suggest further clarification.

We have tried to clarify this bit by adding some text and describing the idea more clearly. The new text reads:

" However, the tool can be used to investigate pathogen outbreak and emergence potential under different pathogen range expansion scenarios. That is, by simulating pathogen spread into new geographic locations, the model could be used to identify potential pathogens that may also shift their geographic ranges. Since the method is based on dissimilarity of countries and pathogen distributions, it is possible to examine the expected outcome as pathogen distributions become more (or less) homogeneous, or countries become more (or less) dissimilar in their pathogen communities. Finally, the sensitivity of the model to pathogen shifts, transients, or imperfect detection are all things that can be tested using simulation approaches, and may provide further insight into our ability to forecast outbreak and emergence events."

P17 L380 – “Though this casts doubt on biogeographic tools like “co-zones” as standalone tools for surveillance or outbreak response, our study is a compelling indicator that community data could be very easily leveraged alongside other socioecological predictors to forecast disease emergence as an ecosystem process rather than a single-species one.” – No ref for “co-zones” but as above, if you are referring to Murray et al., 2015 I don’t think I implied that this should be a stand-alone tool but rather an approach that could provide some of the earliest insights or prior expectations in data poor situations for surveillance or outbreak response.

Your study provides further evidence to support this, at least for outbreaks and re-emergence, but also highlights something of a prediction ceiling so to speak for local emergences and long time horizons. So again this sentence seems a bit misplaced or caricatured. It rather appears we make the same basic points about the potential to leverage community data more effectively.

We apologize for the lack of clarity. We do feel it is important to maintain a critical eye towards the efficacy of models, as our paper attempts to set the outer limits of model accuracy (and, ultimately, draws them in a slightly different place than Murray et al.; we would not recommend the use of our models, for example, during an Ebola outbreak). However, the use of the word “standalone” created a misleading impression of Murray’s paper and of our own intentions. We have rephrased, and removed comparison to Murray’s paper:

“For this definition of emergence (and this dataset), we suggest this indicates an outer bound of predictive capacity based *only* on community structure; however, community data could be very easily leveraged alongside other socioecological predictors to forecast disease emergence as an ecosystem process rather than a single-species one.”

Also, I think there could be a useful distinction with these approaches between the ‘where’ and ‘when’ of

prediction that could be better drawn out. Historical co-occurrence has clear predictive value for where things might turn up in future based on where they have turned up in the past but a much tougher question, as also touched upon here, is when it will turn up.

We agree this is an important question, but our model explicitly avoids predicting temporal trends, and we believe this is beyond the scope of this manuscript; we hope future work will expand on this question more.

I hope some of these thoughts will be useful for your revisions.

We thank the reviewer for their thoughtful comments, which we believe have improved the current manuscript.

Kris Murray